mathematical modelling/computer modelling and simulation

SARS-CoV-2, shutdowns, agent-based model, Canada, non-pharmaceutical interventions, infectious disease modelling

**Author for correspondence:**
Victoria Ng
e-mail: victoria.ng@canada.ca

# Modelling the impact of shutdowns on resurging SARS-CoV-2 transmission in Canada

Victoria Ng, Aamir Fazil, Lisa A. Waddell,

Patricia Turgeon, Ainsley Otten and Nicholas H. Ogden

Public Health Risk Sciences Division, National Microbiology Laboratory, Public Health Agency of Canada, Guelph, Ontario and St Hyacinthe, Quebec, Canada

VN, 0000-0002-7619-541X; NHO, 0000-0002-1062-7283

Background: Shutdowns are enacted when alternative public health measures are insufficient to control the epidemic and the population is largely susceptible. An age-stratified agent-based model was developed to explore the impact of shutdowns to control SARS-CoV-2 transmission in Canada under the assumption that current efforts to control the epidemic remains insufficient and in the absence of a vaccine. Methods: We estimated the current levels of interventions in Canada to generate a baseline scenario from 7 February to 7 September 2020. Four aspects of shutdowns were explored in scenarios that ran from 8 September 2020 to 7 January 2022, these included the impact of how quickly shutdowns are implemented, the duration of shutdowns, the minimum break (delays) between shutdowns and the types of sectors to shutdown. Comparisons among scenarios were made using cases, hospitalizations, deaths and shutdown days during the 700-day model runs. Results: We found a negative relationship between reducing SARS-CoV-2 transmission and the number of shutdown days. However, we also found that for shutdowns to be optimally effective, they need to be implemented fast with minimal delay, initiated when community transmission is low, sustained for an adequate period and be stringent and target multiple sectors, particularly those driving transmission. By applying shutdowns in this manner, the total number of shutdown days could be reduced compared to delaying the shutdowns until further into the epidemic when transmission is higher and/or implementing short insufficient shutdowns that would require frequent re-implementation. This paper contrasts a range of shutdown strategies and trade-offs between health outcomes and economic metrics that need to be considered within the local context. Interpretation: Given the immense socioeconomic impact of shutdowns, they should be avoided where possible

and used only when other public health measures are insufficient to control the epidemic. If used, the time it buys to delay the epidemic should be used to enhance other equally effective, but less disruptive, public health measures.

# 1. Introduction

As coronavirus disease (COVID-19) continues to spread globally with the resurgence of cases in many countries, multiple public health interventions that were employed during the initial months of the epidemic are being re-implemented [1]. A key intervention that contributed to the control of the initial epidemic was unprecedented restrictive shutdowns that limited person-to-person contacts. In Canada, these shutdowns included school closures, limiting public and personal gatherings, and shutting down non-essential businesses [2]. While shutdowns are one of the most disruptive measures to be implemented with widespread socioeconomic impacts, they are also the most effective non-pharmaceutical measures for controlling the exponential spread of severe acute respiratory syndrome coronavirus 2 (SARS-CoV-2) [3,4].

Studies have projected that when restrictive closures are lifted in Canada, if alternative public health measures such as detection and isolation of cases, and tracing and quarantining of close contacts of cases are insufficient, subsequent waves of infections will occur [5–7]. These modelled projections have unfolded in reality in recent months across Canada as many provinces struggle to regain control of the epidemic [8]. As of August 2020, only 1% of the Canadian population is estimated to have been infected with SARS-CoV-2 [9]. December 2020 marked the arrival of multiple vaccines on the global market with the first vaccine approved, procured and administered in Canada [10]. While vaccines offer hope that restrictive shutdowns will be temporary, because it will take most of 2021 to vaccinate the population to reach herd immunity, the high proportion of the population that will remain susceptible in 2021 will make the control of the epidemic difficult without the use of restrictive shutdowns to complement other less disruptive measures. Further, the recent emergence of a mutated strain of SARS-CoV-2 with 40–70% greater person-to-person transmission than pre-existing variants indicate strict measures will need to be imposed in countries experiencing a resurgence in order to regain control [11]. However, shutdowns cannot be implemented indefinitely given the immense socioeconomic cost and should be used by governments to keep healthcare systems at capacity and to build capacity for alternative public health measures.

In this study, we used an age-stratified agent-based model developed for the Canadian population to explore restrictive shutdowns to control the epidemic when alternative public health measures are insufficient. We focused on specific aspects of shutting down including the impact of speed, duration, delays and types of shutdowns on COVID-19 transmission.

# 2. Methods

The methods describing the agent-based model used in this study have been published [5]. Briefly, the model is an age-stratified agent-based simulation model for SARS-CoV-2 transmission in Canada. The model is a simplified approximation of the Canadian situation and does not consider the details of varying geographical dispersion of cases seen across provinces and territories. Technical details of the model including parameters derived from published studies and Canadian data are presented in the electronic supplementary material. The definition for controlling an epidemic in this study was to prevent the epidemic from exponential growth and exceeding a clinical attack rate of over 10% over the 2-year model period (February 2020 to January 2022).

## 2.1. Canadian baseline and estimating current levels of interventions

Four interventions were applied in the baseline; these include case detection and isolation, contact tracing and quarantine, closures and personal physical distancing (electronic supplementary material, table S8), representing the Canadian situation from 7 February to 7 September 2020 (figure 1). During the baseline period, we assume approximately 20% of symptomatic cases are detected via case testing and isolated for the remainder of their infectious period. This was estimated from studies on Canadian under-reporting showing approximately 20–25% of cases are detected [12–14] with around 75% of identified cases

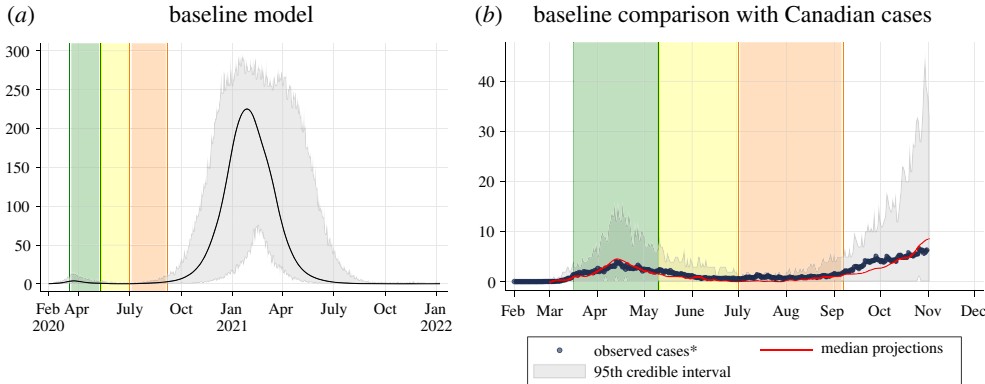

**Figure 1.** Baseline scenario. (*a*) Projected epidemic curve showing daily clinical incident cases per 100 000 people for the baseline model. The black line represents the median value across 50 model realizations. The grey area represents the 95% credible interval across 50 model realizations. The green bar represents 16 March to 10 May (initial shutdown), the yellow bar represents 11 May to 30 June (easing of shutdown) and the orange bar represents 1 July to 7 September (changes in summer). (*b*) Comparison between locally acquired Canadian cases by onset date (blue dots) and the median modelled clinical cases from the baseline model from 50 model realizations (red line). The grey area represents the 95% credible interval across 50 model realizations. Note that the baseline model was fitted to Canadian levels of interventions up until 8 September (the end of the orange bar) and does not consider additional interventions occurring across the country beyond 8 September.

voluntarily self-isolating [15]. We assume 50% of contacts with detected cases are traced and quarantined and when cases reach 50 active cases per 100 000, contact tracing ceases based on the threshold at which Canadian jurisdictions have been overwhelmed within contact tracing capacity [16]. Restrictive closures are implemented in three phases from 16 March to 7 September 2020: 100% of schools, 50% workplaces (teleworkers) and 50% mixed-age venues (non-essential businesses and shared public facilities) are closed in phase 1 (16 March to 10 May), 100% of schools remain closed and 40% of individuals continue to telework and non-essential businesses and shared public facilities reopen in phase 2 (11 May to 30 June), 65% of schools in the model reopen to represent summer camps and activities bringing children together over the summer and 36% of individuals continue to telework in phase 3 (1 July to 7 September). Changes in closures were modelled based on the decline in mobility observed during the corresponding time periods using Google mobility data and Statistics Canada's surveys [17,18]. Personal physical distancing was also introduced in three phases with changes in contact rate and compliance from 16 March to 7 September 2020: 30% contact rate reduction with 100% compliance when outside of the household during the initial shutdown in phase 1 (15 March to 10 May) reflecting precautions Canadians were taking during the initial shutdown [19], compliance remained at 100% but contact rates were reduced to 80% when outside the household to reflect the gradual lifting of closures but continued adherence to physical distancing in phase 2 (11 May to 30 June) [20,21], contact rates were reduced to only 55% with compliance shifting by age group in phase 3 (1 July 2020 to model end): 0–4 (33%), 5–9 (33%), 11–14 (33%), 15–19 (33%), adult1 (50%), adult2 (75%), adult3 (85%), senior1 (90%), senior2 (95%), elderly (95%) [20,22]. Changes to contact rate reduction and compliance were estimated from Canadian survey data on behavioural variations to physical distancing over time [20–23] and the observed changes in COVID-19 cases by age groups over time [24]. In the baseline scenario, we assume everything reopens on 8 September 2020; which in reality is not the case as a large proportion of Canadians continue to telework [17] and subsequent shutdowns have occurred [1,2]. This was intentional to explore the impact of further shutdowns. We also assume the last phase of physical distancing contact rate reduction and compliance continues for the remaining model run. Last, the baseline assumes an importation rate of one case per 100 000 per week estimated from internal modelling (Public Health Agency of Canada Modelling group biweekly report (2021)). Additional information on the baseline including model fitting is in the electronic supplementary material.

## 2.2. Shutdown scenarios

Building on the baseline, the model explored four shutdown scenarios separately to explore the impact of further shutdowns on the resurgence of SARS-CoV-2 transmission in Canada. The scenarios explored shutdowns occurring from 8 September 2020 (day 214) to the model end (7 January 2022, day 700); a period of 487 days (table 1). Shutdowns, unless otherwise stated, included the closure of 100% of

**Table 1.** Summary of shutdown scenarios and the models (bolded) explored (18 models in total).

| changes that could be applied from 8 September 2020 onward | Scenario 1: speed of shutdowns | Scenario 2: duration of shutdowns | Scenario 3: minimum break between subsequent shutdowns | Scenario 4: extensiveness of shutdowns |
|---|---|---|---|---|
| threshold to trigger shutdowns | **A. 50 active cases[a] per 100 000** **B. 100 active cases[a] per 100 000** **C. 150 active cases[a] per 100 000** **D. 200 active cases[a] per 100 000** | 100 active cases[a] per 100 000 (unchanged) | 100 active cases[a] per 100 000 (unchanged) | 100 active cases[a] per 100 000 (unchanged) |
| duration of shutdowns | 42 days (unchanged) | **A. 28 days (4 weeks)** **B. 42 days (6 weeks)** **C. 56 days (8 weeks)** **D. 70 days (10 weeks)** | 42 days (unchanged) | 42 days (unchanged) |
| duration between shutdowns (minimum break between shutdowns) | none, subsequent shutdowns occur as soon as trigger for shutting down is reached (unchanged) | none, subsequent shutdowns occur as soon as trigger for shutting down is reached (unchanged) | **A. 28 days (4 weeks)** **B. 56 days (8 weeks)** **C. 84 days (12 weeks)** **D. 112 days (16 weeks)** | none, subsequent shutdowns occur as soon as trigger for shutting down is reached (unchanged) |
| sectors that are closed during shutdowns | 100% schools, 50% workplaces and 50% mixed-age venues for each shutdown (unchanged) | 100% schools, 50% workplaces and 50% mixed-age venues for each shutdown (unchanged) | 100% schools, 50% workplaces and 50% mixed-age venues for each shutdown (unchanged) | **A. Baseline—no further shutdowns** **B. All (50% workplaces, 100% schools and 50% mixed-age venues)** **C. 50% workplaces only** **D. 100% schools only** **E. 50% mixed-age venues only** **F. 50% mixed-age venues and 50% workplaces together** |

[a]Active cases in the model represented the total number of cases that were symptomatic or hospitalized on a given day. It did not include asymptomatic cases. On average, mild cases were active for 6.5 days (range of 5–12 days), hospitalized cases were active for 14.5 days (range of 7–21 days) and ICU-admitted cases were active for 23 days (range of 12.5–33.5 days). Cases were considered resolved when they are no longer symptomatic (mild cases), discharged from hospital or had died.

schools, 50% workplaces (teleworkers) and 50% mixed-age venues (non-essential businesses and shared public facilities) [17]. Shutdown scenarios were developed based on realistic Canadian targets according to previous and current shutdown efforts and shutdown strategies implemented globally [1,2,17].

Scenario 1 (S1) explored the speed of shutdowns when the following triggers were reached, reflecting some Canadian thresholds that have been set to trigger shutdowns [24–26]: (A) 50 active cases per 100 000 (fast response), (B) 100 active cases per 100 000, (C) 150 active cases per 100 000 and (D) 200 active cases per 100 000 (slow response). Active cases in the model represented the total number of cases that were symptomatic or hospitalized on a given day, i.e. clinical cases once symptoms have begun. These thresholds were selected to reflect thresholds used by some provinces in Canada while recognizing some jurisdictions use healthcare system capacity as a trigger for shutdowns, which are also considered jointly with economic impacts. Shutdowns were implemented for 42 days at a time and re-implemented when the threshold for shutting down was reached again with no minimum break in between.

Scenario 2 (S2) explored the duration of shutdowns by shutting down for (A) 28 days, (B) 42 days, (C) 56 days and (D) 70 days at a time with shutdowns triggered when active cases reached 100 per 100 000 with no minimum break between consecutive shutdowns. Shutdowns were released after the corresponding period of shutdown ended regardless of the number of active cases per 100 000 remaining and re-implemented when 100 active cases per 100 000 was reached again.

Scenario 3 (S3) explored the delays between shutdowns by shutting down with a minimum break between consecutive shutdowns. Shutdowns were modelled with a minimum of (A) 28 days between shutdowns; (B) 56 days between shutdowns; (C) 84 days between shutdowns and (D) 112 days between shutdowns. Shutdowns were triggered when active cases reached 100 cases per 100 000 and the minimum time between shutdowns had lapsed regardless of the number of active cases per 100 000 remaining.

Scenario 4 (S4) explored the extensiveness of shutdowns by shutting down different sectors individually and in combination with each other: (A) Canadian baseline, no further shutdown, (B) all shutdown (100% schools, 50% workplaces and 50% mixed-age venues), (C) 50% workplaces only (teleworkers), (D) 100% schools only, (E) 50% mixed-age venues only (nonessential businesses and shared public facilities) and (F) 50% mixed-age venues and 50% workplaces together. Shutdowns were modelled on the decline in mobility observed during March to May [17]. Shutdowns were triggered at 100 active cases per 100 000, imposed for 42 days at a time with no minimum break between subsequent shutdowns.

## 2.3. Model output

The model produced daily cumulative and incident counts for each age group for each health state. The stochastic outputs are presented as medians from 50 realizations (simulations) per scenario and values from the 2.5th and 97.5th percentiles are presented as the 95% credible interval (CI). The model was developed and simulated in AnyLogic. Statistical analyses and graphs were created using StataCorp 2019, Stata Statistical Software: Release 16, College Station, TX: StataCorp LLC.

# 3. Results

Results on the total attack rate, clinical attack rate, asymptomatic attack rate, total hospitalizations (acute and intensive care unit (ICU)), acute hospital admissions, ICU admissions and mortality rate by age groups for each scenario and model (16 in total) are presented in the electronic supplementary material, table. Summary tables on key model outputs for scenarios are presented in tables 2–6. Figures present epidemic curves of daily clinical (i.e. symptomatic) incident cases per 100 000 people for each model (figures 2–5).

## 3.1. Scenario 1: speed of shutdowns

When current public health measures are insufficient to control an epidemic, the faster to shutdown (50 active cases per 100 000), the lower the number of cases, hospitalizations and deaths, but the higher the number of shutdown days due to shutting down more often (table 2 and figure 2). Earlier shutdowns prevent more cases, hospitalizations and deaths (2.56% clinical attack rate when shutting down at 50 active cases per 100 000; 4.55% at 100 active cases per 100 000; 6.33% at 150 active cases per 100 000

**Table 2.** Summary of model outputs for Scenario 1: speed of shutdowns. Median values from 50 realizations are presented in the table with 2.5th percentile and 97.5th percentile values representing the 95% credible interval presented in brackets. The median values indicate the most likely outcome out of 50 realizations.

| Scenario 1: speed of shutdowns | A. 50 active cases per 100 000 | B. 100 active cases per 100 000 | C. 150 active cases per 100 000 | D. 200 active cases per 100 000 |
|---|---|---|---|---|
| total attack rate (%) | 4.25 (3.44–12.79) | 7.51 (6.22–8.58) | 10.39 (8.84–11.87) | 12.95 (10.62–14.38) |
| clinical attack rate (%) | 2.56 (2.11–7.72) | 4.55 (3.79–5.27) | 6.33 (5.43–7.32) | 7.88 (6.45–8.76) |
| asymptomatic attack rate (%) | 1.66 (1.33–5.07) | 2.96 (2.43–3.33) | 4.03 (3.42–4.61) | 5.07 (4.14–5.54) |
| case fatality rate (%) | 1.47 (1.07–2.24) | 1.43 (1.09–1.71) | 1.39 (1.15–1.66) | 1.49 (1.12–1.79) |
| infection fatality rate (%) | 0.90 (0.65–1.38) | 0.88 (0.66–1.04) | 0.84 (0.70–1.01) | 0.90 (0.68–1.08) |
| number of shutdown days (from 8 Sep 2020 to 7 Jan 2022) | 255 (212–297) | 216 (187–257) | 213 (171–253) | 194 (168–214) |
| total cases (clinical and asymptomatic) per 100 000 | 4256 (3444–12 799) | 7517 (6227–8593) | 10 396 (8851–11 877) | 12 961 (10 635–14 391) |
| total clinical cases per 100 000 | 2565 (2110–7727) | 4554 (3795–5277) | 6338 (5432–7326) | 7887 (6458–8772) |
| total asymptomatic cases per 100 000 | 1662 (1334–5072) | 2965 (2433–3331) | 4037 (3419–4610) | 5073 (4443–5545) |
| total hospitalized cases (acute and ICU) per 100 000 | 266 (217–787) | 470 (388–540) | 645 (42–762) | 782 (631–904) |
| total acute hospitalized cases per 100 000 | 198 (165–577) | 350 (279–410) | 478 (94–563) | 582 (465–681) |
| total cases admitted into the ICU per 100 000 | 68 (48–209) | 119 (91–140) | 159 (29–197) | 197 (155–230) |
| total deaths per 100 000 | 36 (26–173) | 65 (47–82) | 88 (70–115) | 116 (82–146) |
| proportion of infections acquired at school (%) | 5.78 (4.35–7.55) | 6.31 (5.25–7.52) | 6.76 (5.66–7.78) | 7.03 (6.22–8.18) |
| proportion of infections acquired at work (%) | 11.06 (9.77–12.22) | 11.34 (10.62–12.52) | 11.39 (10.55–12.48) | 11.46 (10.98–12.13) |
| proportion of infections acquired in mixed-age venues (%) | 21.90 (20.47–23.75) | 22.03 (20.52–23.15) | 22.09 (21.21–23.27) | 21.85 (20.95–22.93) |
| proportion of infections acquired in the household (%) | 61.20 (59.47–62.15) | 60.23 (59.59–60.95) | 59.79 (58.99–60.63) | 59.53 (58.70–59.89) |

**Table 3.** Summary of model outputs for Scenario 2: duration of shutdowns. Median values from 50 realizations are presented in the table with 2.5th percentile and 97.5th percentile values representing the 95% credible interval presented in brackets. The median values indicate the most likely outcome out of 50 realizations.

| Scenario 2: duration of shutdowns | A. 28-day (4 weeks) shutdown | B. 42-day (6 weeks) shutdown | C. 56-day (8 weeks) shutdown | D. 70-day (10 weeks) shutdown |
|---|---|---|---|---|
| total attack rate (%) | 8.45 (7.29–9.98) | 7.57 (6.63–8.36) | 6.93 (5.62–7.92) | 6.33 (4.86–7.45) |
| clinical attack rate (%) | 5.17 (4.52–6.14) | 4.65 (4.08–5.06) | 4.22 (3.50–4.84) | 3.86 (2.97–4.61) |
| asymptomatic attack rate (%) | 3.26 (2.78–3.83) | 2.94 (2.54–3.30) | 2.66 (2.12–3.09) | 2.47 (1.88–2.86) |
| case fatality rate (%) | 1.40 (1.11–1.69) | 1.39 (1.14–1.81) | 1.39 (1.15–1.78) | 1.43 (1.01–1.73) |
| infection fatality rate (%) | 0.85 (0.66–1.04) | 0.85 (0.70–1.10) | 0.85 (0.71–1.09) | 0.87 (0.62–1.06) |
| number of shutdown days (from 8 Sep 2020 to 7 Jan 2022) | 211 (171–256) | 218 (185–256) | 229 (224–283) | 256 (211–283) |
| total cases (clinical and asymptomatic) per 100 000 | 8461 (7298–9991) | 7580 (6633–8368) | 6933 (5623–7924) | 6334 (4864–7457) |
| total clinical cases per 100 000 | 5180 (4528–6155) | 4652 (4081–5063) | 4227 (3499–4844) | 3867 (2976–4617) |
| total asymptomatic cases per 100 000 | 3271 (2791–3836) | 2943 (2540–3305) | 2662 (2124–3090) | 2474 (1883–2863) |
| total hospitalized cases (acute and ICU) per 100 000 | 520 (462–648) | 466 (391–538) | 431 (355–492) | 391 (297–468) |
| total acute hospitalized cases per 100 000 | 383 (343–475) | 341 (288–412) | 317 (259–367) | 291 (219–348) |
| total cases admitted into the ICU per 100 000 | 135 (115–168) | 117 (95–141) | 108 (80–133) | 95 (69–123) |
| total deaths per 100 000 | 74 (57–90) | 65 (50–85) | 59 (40–74) | 53 (38–70) |
| proportion of infections acquired at school (%) | 6.38 (5.31–7.75) | 6.32 (5.29–7.15) | 6.26 (5.17–7.31) | 6.09 (4.70–7.56) |
| proportion of infections acquired at work (%) | 11.49 (10.42–12.30) | 11.47 (10.37–12.85) | 11.24 (10.05–12.39) | 11.28 (10.31–12.56) |
| proportion of infections acquired in mixed-age venues (%) | 22.17 (21.03–23.49) | 21.95 (20.53–23.02) | 22.13 (20.97–23.33) | 21.97 (20.43–23.45) |
| proportion of infections acquired in the household (%) | 60.12 (59.24–60.73) | 60.24 (59.26–61.10) | 60.48 (59.63–61.13) | 60.58 (59.80–61.44) |

**Table 4.** Summary of model outputs for Scenario 3: minimum break between subsequent shutdowns. Median values from 50 realizations are presented in the table with 2.5th percentile and 97.5th percentile values representing the 95% credible interval presented in brackets. The median values indicate the most likely outcome out of 50 realizations.

| Scenario 3: minimum break between subsequent shutdowns | A. 28 days (4 weeks) break between shutdowns | B. 56 days (8 weeks) break between shutdowns | C. 84 days (12 weeks) break between shutdowns | D. 112 days (16 weeks) break between shutdowns |
|---|---|---|---|---|
| total attack rate (%) | 7.84 (6.10–9.78) | 15.38 (9.33–18.49) | 21.77 (17.58–24.95) | 25.28 (21.93–29.07) |
| clinical attack rate (%) | 4.78 (3.10–5.98) | 9.36 (5.65–11.23) | 13.24 (10.73–15.23) | 15.45 (13.30–17.53) |
| asymptomatic attack rate (%) | 3.07 (2.10–3.88) | 6.01 (3.68–7.26) | 8.52 (6.85–9.83) | 9.83 (8.66–11.54) |
| case fatality rate (%) | 1.39 (1.10–1.78) | 1.85 (1.34–2.20) | 2.36 (2.06–2.62) | 2.50 (2.11–2.81) |
| infection fatality rate (%) | 0.84 (0.10–1.08) | 1.12 (0.81–1.34) | 1.44 (1.26–1.61) | 1.52 (1.28–1.71) |
| number of shutdown days (from 8 Sep 2020 to 7 Jan 2022) | 233 (172–255) | 186 (169–213) | 128 (127–171) | 127 (85–128) |
| total cases (clinical and asymptomatic) per 100 000 | 7850 (6101–9786) | 15 399 (9335–18 512) | 21 788 (17 597–24 977) | 25 301 (21 950–29 102) |
| total clinical cases per 100 000 | 4787 (3723–5986) | 9374 (5656–11 241) | 13 254 (10 739–15 249) | 15 468 (13 315–17 550) |
| total asymptomatic cases per 100 000 | 3081 (2378–3813) | 6020 (3679–7271) | 8525 (6858–9843) | 9835 (8669–11 552) |
| total hospitalized cases (acute and ICU) per 100 000 | 489 (385–611) | 937 (584–1118) | 1344 (1058–1565) | 1576 (1287–1874) |
| total acute hospitalized cases per 100 000 | 357 (284–447) | 694 (417–849) | 1003 (796–1155) | 1168 (971–1378) |
| total cases admitted into the ICU per 100 000 | 120 (91–164) | 238 (152–278) | 350 (271–409) | 397 (312–501) |
| total deaths per 100 000 | 66 (50–89) | 170 (76–231) | 318 (225–380) | 383 (299–504) |
| proportion of infections acquired at school (%) | 6.21 (5.40–7.54) | 7.22 (6.36–7.94) | 8.07 (7.44–8.70) | 8.85 (8.20–9.38) |
| proportion of infections acquired at work (%) | 11.37 (10.11–12.40) | 11.45 (10.65–12.06) | 11.54 (10.80–12.19) | 11.60 (10.95–12.12) |
| proportion of infections acquired in mixed-age venues (%) | 22.06 (20.68–23.24) | 22.02 (20.90–22.66) | 21.66 (21.06–22.55) | 21.52 (20.92–22.25) |
| proportion of infections acquired in the household (%) | 60.35 (59.65–60.96) | 59.37 (58.70–59.88) | 58.62 (58.14–59.20) | 58.06 (57.23–58.54) |

**Table 5.** Summary of model outputs for Scenario 4: extensiveness of shutdowns (part 1). Median values from 50 realizations are presented in the table with 2.5th percentile and 97.5th percentile values representing the 95% credible interval presented in brackets. The median values indicate the most likely outcome out of 50 realizations.

| Scenario 4: extensiveness of shutdowns | A. baseline (no further shutdowns) | B. all shutdown (100% schools, 50% workplaces and 50% mixed-age venues) | C. 50% workplaces only |
|---|---|---|---|
| total attack rate (%) | 41.11 (40.12–42.22) | 6.50 (5.33–7.59) | 33.59 (32.38–34.67) |
| clinical attack rate (%) | 25.13 (24.38–25.74) | 3.97 (3.25–4.60) | 20.36 (19.29–21.12) |
| asymptomatic attack rate (%) | 16.00 (15.64–16.47) | 2.55 (2.12–2.99) | 13.20 (12.19–13.69) |
| case fatality rate (%) | 2.96 (2.74–3.12) | 1.40 (1.15–1.73) | 2.84 (2.2–3.04) |
| infection fatality rate (%) | 1.80 (1.67–1.90) | 0.84 (0.70–1.04) | 1.72 (1.1–1.84) |
| number of shutdown days (from 8 Sep 2020 to 7 Jan 2022) | 0 (0–0) | 195 (168–213) | 255 (213–255) |
| total cases (clinical and asymptomatic) per 100 000 | 41 145 (40 157–42 256) | 6503 (5336–7595) | 33 625 (32 309–34 703) |
| total clinical cases per 100 000 | 25 151 (24 405–25 766) | 3973 (3249–4607) | 20 384 (19 505–21 137) |
| total asymptomatic cases per 100 000 | 16 015 (15 659–16 490) | 2556 (2124–2988) | 13 217 (12 804–13 707) |
| total hospitalized cases (acute and ICU) per 100 000 | 2618 (2483–2763) | 408 (324–464) | 2091 (1985–2242) |
| total acute hospitalized cases per 100 000 | 1957 (1859–2073) | 292 (239–351) | 1568 (1477–1671) |
| total cases admitted into the ICU per 100 000 | 664 (615–716) | 101 (76–132) | 532 (479–574) |
| total deaths per 100 000 | 740 (670–790) | 54 (45–71) | 577 (542–623) |
| proportion of infections acquired at school (%) | 10.23 (9.92–10.68) | 7.13 (6.13–8.22) | 11.66 (11.10–11.98) |
| proportion of infections acquired at work (%) | 12.26 (11.83–12.66) | 10.98 (9.84–11.76) | 7.21 (6.75–7.78) |
| proportion of infections acquired in mixed-age venues (%) | 21.96 (21.50–22.23) | 21.45 (19.69–22.76) | 23.94 (23.57–24.48) |
| proportion of infections acquired in the household (%) | 55.56 (55.29–56.02) | 60.53 (59.63–61.40) | 57.19 (56.87–57.43) |

**Table 6.** Summary of model outputs for Scenario 4: extensiveness of shutdowns (part 2). Median values from 50 realizations are presented in the table with 2.5th percentile and 97.5th percentile values representing the 95% credible interval presented in brackets. The median values indicate the most likely outcome out of 50 realizations.

| Scenario 4: extensiveness of shutdowns | D. 100% schools only | E. 50% mixed-age venues only | F. 50% mixed-age venues and 50% workplaces |
|---|---|---|---|
| total attack rate (%) | 21.47 (18.90–23.50) | 10.75 (9.40–13.30) | 9.10 (7.80–10.36) |
| clinical attack rate (%) | 13.31 (11.68–14.59) | 6.49 (5.65–8.01) | 5.45 (4.71–6.24) |
| asymptomatic attack rate (%) | 8.17 (7.22–8.97) | 4.27 (3.72–5.29) | 3.61 (3.05–4.12) |
| case fatality rate (%) | 2.44 (2.16–2.73) | 1.36 (1.07–1.65) | 1.36 (1.08–1.72) |
| infection fatality rate (%) | 1.51 (1.33–1.68) | 0.81 (0.65–0.99) | 0.82 (0.65–1.04) |
| number of shutdown days (from 8 Sep 2020 to 7 Jan 2022) | 340 (297–383) | 339 (288–376) | 297 (255–339) |
| total cases (clinical and asymptomatic) per 100 000 | 21 492 (18 923–23 519) | 10 762 (9409–13 309) | 9109 (7810–10 371) |
| total clinical cases per 100 000 | 13 320 (11 693–14 602) | 6501 (5656–8013) | 5457 (4714–6248) |
| total asymptomatic cases per 100 000 | 8173 (7230–8983) | 4277 (3719–5296) | 3619 (3055–4123) |
| total hospitalized cases (acute and ICU) per 100 000 | 1412 (1240–1598) | 634 (548–754) | 539 (452–637) |
| total acute hospitalized cases per 100 000 | 1062 (949–1179) | 475 (400–554) | 396 (343–455) |
| total cases admitted into the ICU per 100 000 | 354 (301–423) | 158 (134–199) | 131 (106–166) |
| total deaths per 100 000 | 328 (260–383) | 89 (67–119) | 74 (57–95) |
| proportion of infections acquired at school (%) | 1.41 (0.95–1.91) | 11.39 (10.14–12.20) | 12.38 (10.95–14.06) |
| proportion of infections acquired at work (%) | 15.19 (14.61–15.97) | 11.22 (10.41–12.29) | 8.47 (7.49–9.63) |
| proportion of infections acquired in mixed-age venues (%) | 25.90 (25.26–26.95) | 17.23 (16.14–18.56) | 18.52 (17.41–19.82) |
| proportion of infections acquired in the household (%) | 57.43 (57.01–57.87) | 60.13 (59.57–60.74) | 60.60 (59.98–61.09) |

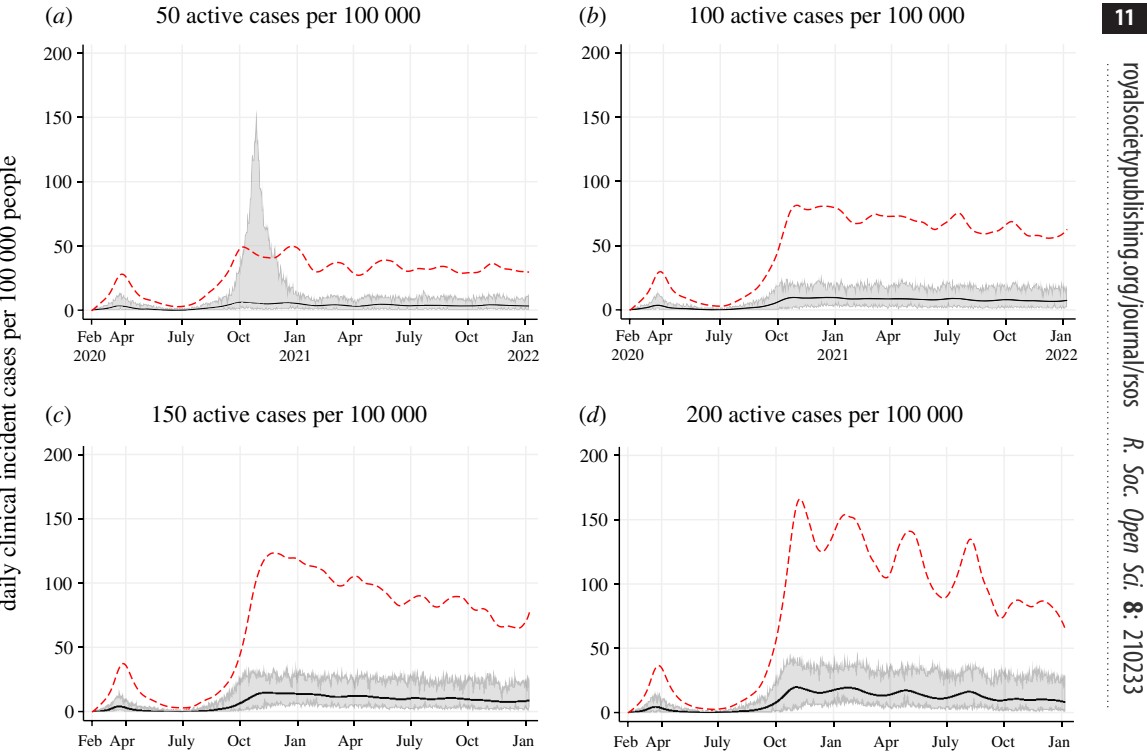

**Figure 2.** Projected epidemic curves showing daily clinical incident cases per 100 000 people for Scenario 1 at different thresholds to trigger shutdowns: 50 active cases per 100 000 (*a*); 100 active cases per 100 000 (*b*); 150 active cases per 100 000 (*c*) and 200 active cases per 100 000 (*d*). The black line represents the median clinical incident value across 50 model realizations. The shaded grey area represents the 95% credible interval. The red dash line represents the median of prevalent active clinical cases used as the threshold for shutting down.

and 7.88% at 200 active cases per 100 000). However, they require an increasing number of shutdowns days (255 out of a possible 487 days when shutting down at 50 active cases per 100 000; 216 days at 100 active cases per 100 000; 213 days at 150 active cases per 100 000 and 194 days at 200 active cases per 100 000). Prevalent cases are on average three-fold higher at the beginning of the epidemic when shutting down at 200 active cases compared to shutting down at 50 active cases per 100 000. This difference gradually drops to two-fold as the immunity level increases in the higher threshold scenarios (figure 2). In a few model runs when shutting down at 50 active cases per 100 000 (figure 2*a*), the shutdown impact was not always immediate, resulting in a larger epidemic peak despite shutdown being implemented (though this was notably smaller compared to the baseline—figure 1*a*). This phenomenon was also observed in scenarios when shutting down at 100 active cases per 100 000 (not visible as the few extreme runs are not captured in the 95% CIs) and occasionally at 150 or 200 active cases per 100 000. This is due to shutdowns having a direct impact on inter-household transmission but not within-household transmission; the latter will continue when shutdowns are implemented because household members do not physically distance from each other. Therefore, at 50 active cases per 100 000, a larger proportion of household members remain susceptible to infection, whereas at 150 or 200 active cases per 100 000, many household members have succumbed to infection and there is a high level of immunity at the population level.

## 3.2. Scenario 2: duration of shutdowns

When public health interventions are insufficient, shorter shutdowns are triggered more frequently: seven complete cycles of shutdowns were observed for the 28-day shutdown model; five cycles of shutdowns for the 42-day model; four cycles of shutdowns for the 56-day model and three to four cycles of shutdowns for the 70-day model when the trigger for shutting down was set to 100 active cases per 100 000 (figure 3). These cycles are observed in both the median incident and prevalent clinical cases. The total number of shutdown days were slightly higher the longer the shutdown period: for the 70-day shutdown the total number of shutdown days was 256, and 229 days, 218 days,

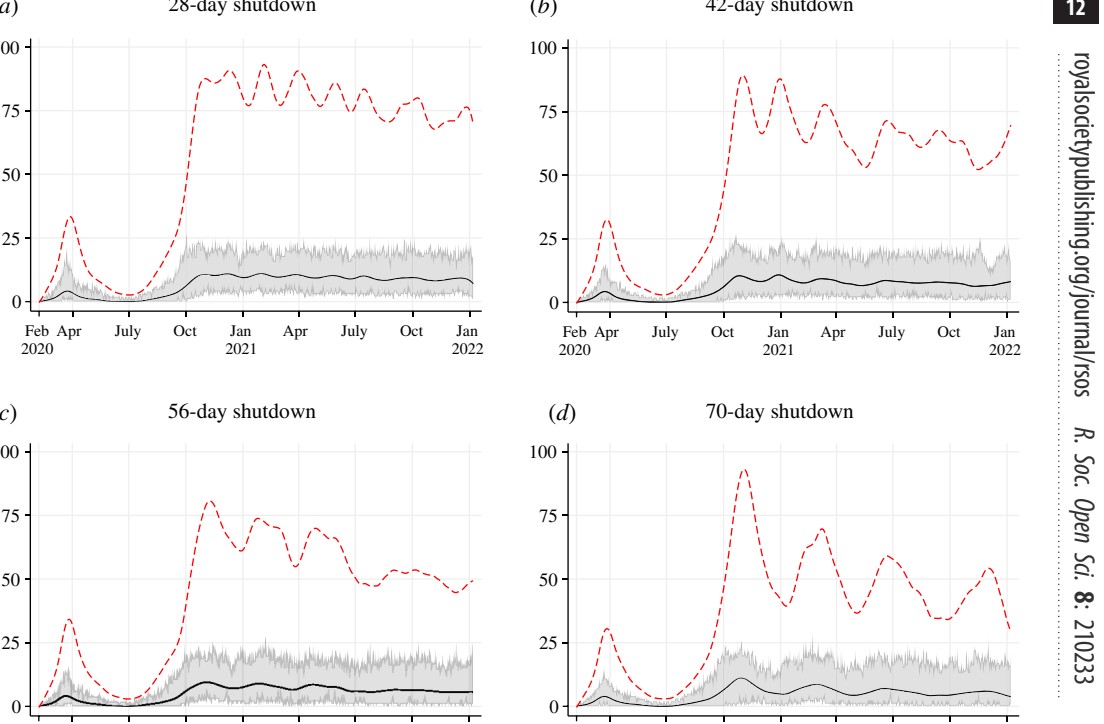

**Figure 3.** Projected epidemic curves showing daily clinical incident cases per 100 000 people for Scenario 2 at different lengths of shutdowns: 28 days (*a*); 42 days (*b*); 56 days (*c*) and 70 days (*d*). The black line represents the median clinical incident value across 50 model realizations. The shaded grey area represents the 95% credible interval. The red dash line represents the median of prevalent active clinical cases.

211 days when the shutdown duration was set at 56-days, 42-days and 28-days, respectively (table 3). The total clinical attack rate was higher when shutting down for shorter periods across many cycles compared to shutting down for longer periods across fewer cycles (5.17% for seven cycles of 28-day shutdowns; 4.65% for five cycles of 42-day shutdowns; 4.22% for four cycles of 56-day shutdowns and 3.86% for three cycles of 70-day shutdowns). Unlike S1, the initial prevalent clinical case count at which shutdowns were triggered were similar across scenarios in S2. Shorter period of shutdowns, when alternative public health measures are insufficient, resulted in more cases, hospitalizations and deaths compared to sustained periods of shutdowns in S2. Comparison of the duration of shutdowns (S2) and the timing of the shutdown (S1) show a higher impact of timing quick re-implementation of the shutdowns. In all four scenarios, a decrease in prevalent cases is observed, with each subsequent shutdown indicating a gradual increase in immunity level in the population.

## 3.3. Scenario 3: delays between shutdowns

When the population is naive and current measures are insufficient, delays between shutdowns will result in progressively larger epidemics (figure 4). The number of shutdown days was higher when the duration between shutdowns were shorter due to more frequent shutdowns being needed: 233 days (28-day break), 186 days (56-day break), 128 days (84-day break) and 127 days (112-day break). Although it appears the effects of an 84-day break and 112-day break between shutdowns is negligible, further analysis from a longer model run showed that a significantly higher number of shutdown days is needed to control the epidemic in the 84-day break scenario (electronic supplementary material, appendix). The clinical attack rate was considerably lower when the duration between shutdowns were shorter: 4.78% with a 28-day break, 9.36% with a 56-day break, 13.24% with an 84-day break and 15.45% with a 112-day break (table 4). In the 56-day break model, a large proportion of the population remained susceptible after the wave in winter 2021, the subsequent wave in spring 2021 was, therefore, larger (figure 4*b*). In comparison, in the 84-day and 112-day break models, a larger proportion of the population were infected during the winter 2021 wave, thus, the subsequent wave in summer 2021 was smaller despite the same shutdown conditions imposed

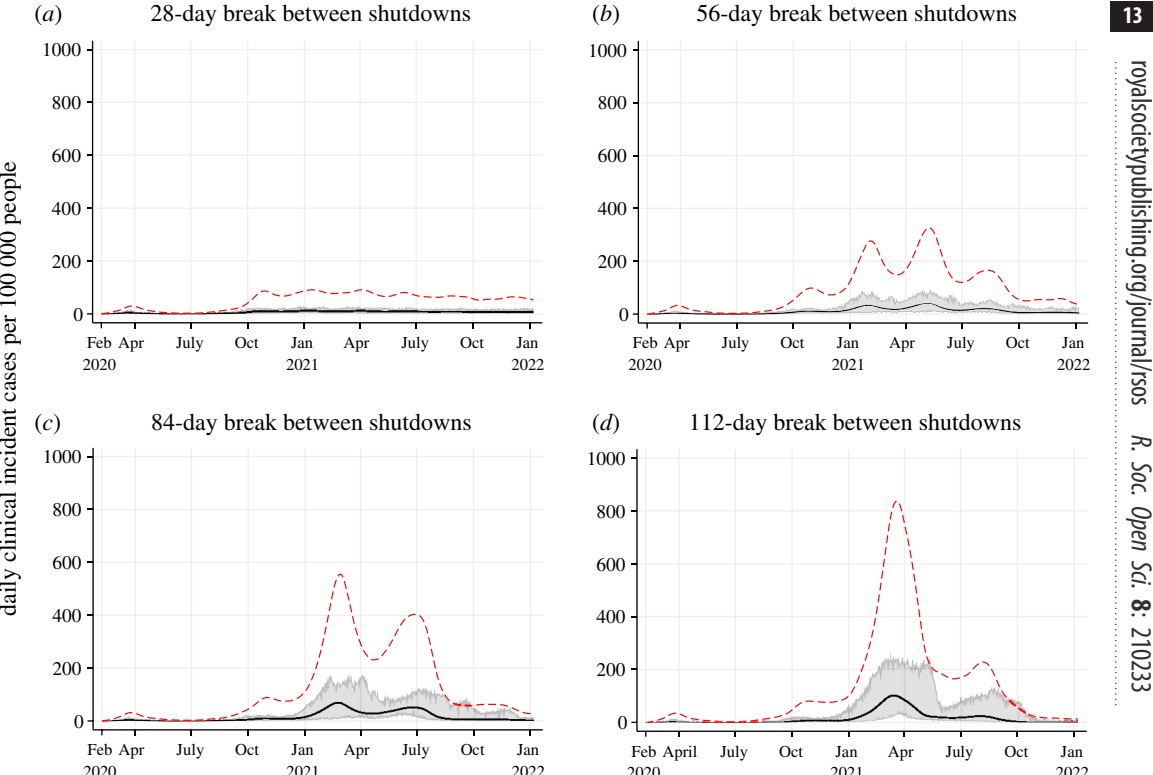

**Figure 4.** Projected epidemic curves showing daily clinical incident cases per 100 000 people for Scenario 3 with different minimum breaks between shutdowns: 28-day break (*a*); 56-day break (*b*); 84-day break (*c*) and 112-day break (*d*). The black line represents the median clinical incident value across 50 model realizations. The shaded grey area represents the 95% credible interval. The red dash line represents the median of prevalent active clinical cases.

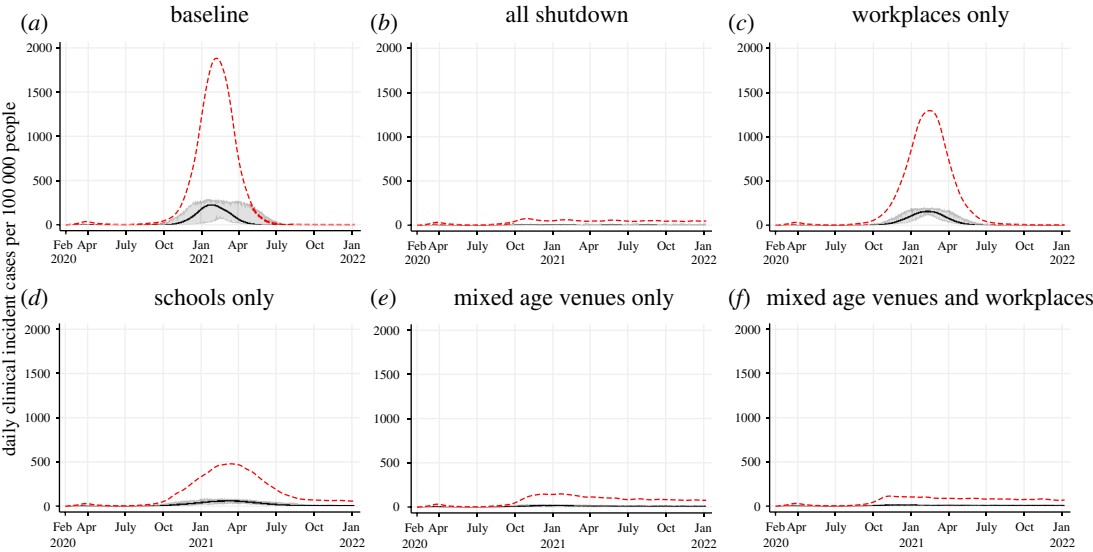

**Figure 5.** Projected epidemic curves showing daily clinical incident cases per 100 000 people for Scenario 4 at varying extensiveness of shutdowns: baseline—no further shutdowns (*a*); all shutdown (100% schools, 50% workplaces and 50% mixed-age venues) (*b*); 50% workplaces only (*c*); 100% schools only (*d*); 50% mixed-age venues only (*e*) and 50% mixed-age venues and 50% workplaces together (*f*). The black line represents the median clinical incident value across 50 model realizations. The shaded grey area represents the 95% credible interval. The red dash line represents the median of prevalent active clinical cases.

(figure 4*c*,*d*). The magnitude of subsequent waves is determined by the proportion of the population that remains susceptible. Delaying a shutdown when a large proportion of the population remains naive can result in up to ten times the number of prevalent clinical cases at peak compared to a no-delay approach (approx. 800 per 100 000 with a 112-day break compared to 80 per 100 000 with a 28-day break), thus

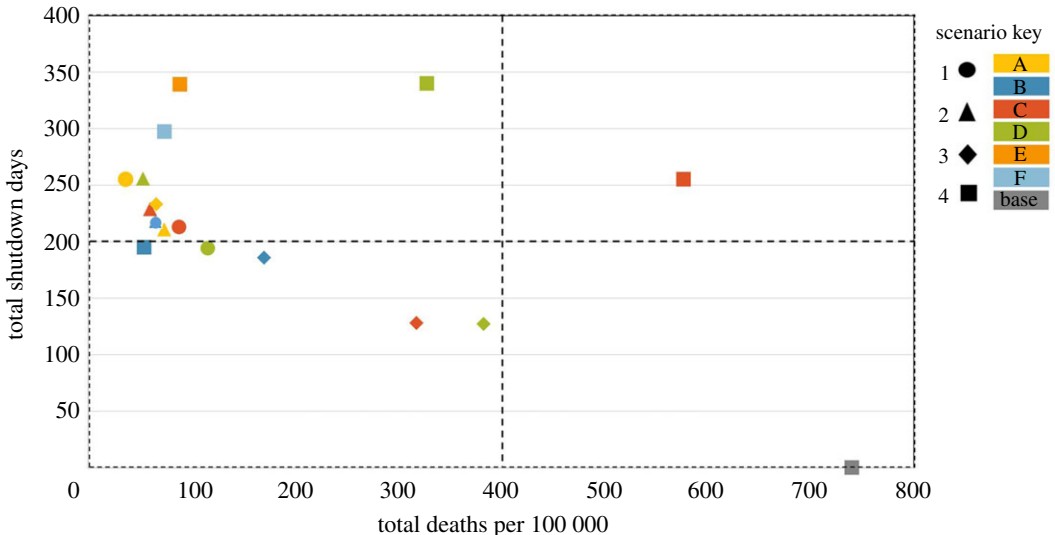

**Figure 6.** The health and economic trade-off presented as the 'price' of a shutdown day versus the number of deaths per 100 000 for each modelled scenario. The lower left quadrant represents the best trade-off in which the total number of deaths per 100 000 is low and the number of shutdown days is close to zero. Circle markers represent scenario 1 models with colours distinguishing models from each other (A—yellow, B—blue, C—red and D—green). Scenario 2 models are represented by triangle markers (A—yellow, B—blue, C—red and D—green). Scenario 3 models are represented by diamond markers (A—yellow, B—blue, C—red and D—green). Scenario 4 models are represented by square markers (A—baseline model in grey, B—blue, C—red, D—green, E—orange and F—light blue). The majority of markers are in the lower and upper left quadrants as these scenarios specifically model shutdowns as a public health measure to reduce deaths, we, therefore, expect the total number of shutdown days to be high and the total deaths per 100 000 to be low in most scenarios.

overwhelming the healthcare system (figure 4). Total hospitalizations increased by three-fold in the 112-day break scenario compared to the 28-day break scenario, while deaths increased by six-fold.

## 3.4. Scenario 4: sectors to shutdown and extensiveness of the shutdown

The clinical attack rate for the baseline scenario with no further shutdown was 25.13% (figure 5*a* and table 5). In comparison, the clinical attack rate for the total shutdown model (50% mixed-age venues, 50% workplaces and 100% schools) was 3.97% (figure 5*b* and table 5). By sector, the closure of 50% workplaces alone was the least effective when exploring shutting down sectors individually (clinical attack rate of 20.36%). The closure of 100% of schools was moderately effective on its own (clinical attack rate of 13.31%) and the most effective was the closure of 50% of mixed-age venues alone (clinical attack rate of 6.49%) (figure 5 and tables 5 and 6). Combining the closure of 50% workplaces with 50% mixed-age venues together improved effectiveness (clinical attack rate of 5.45%) contributing to a 93% reduction in clinical cases compared to the total shutdown model (5.45% versus 3.97%; panel (*f*) versus panel (*b*)). In other words, the additional closure of 100% schools to 50% workplaces and 50% mixed-age venues contributed to a further 7% reduction in clinical cases. The closure of sectors individually contributed to reducing transmission, but contributions were not equal and combining sectors to shut down together were the most effective. Individual closures reduced the proportion of transmission in their respective locations (1.41% in schools when schools were closed, 7.21% in workplaces when workplaces were closed and 17.23% in mixed-age venues when mixed-age venues were closed compared to 10.23%, 12.26% and 21.96% in the baseline; tables 5 and 6) but averted cases from closed locations were not completely prevented due to continued transmission in other locations that remained open, and in the household. Differences in the effectiveness of closures by sector is driven by the community and household structure of the population, age demographics of the population and the types of contacts they have, and the proportion of sectors that were shut down.

## 3.5. Comparison across scenarios

In general, shutdowns are effective in reducing transmission and all models reduced cases, hospitalizations and deaths compared to the baseline. To contrast the different scenarios by deaths per 100 000 and number of shutdown days, figure 6 shows the health and economic trade-off as the 'price'

of a shutdown day in deaths per 100 000 for each modelled scenario. Markers on the lower left quadrant represents the best overall scenario; these represent scenarios with the lowest number of shutdown days combined with the lowest number of deaths. Of the scenarios modelled, only five were found within the lower left quadrant; one scenario (S4, model B — total shutdown) had 54 deaths per 100 000 with a trade-off of 195 shutdown days. The majority of scenarios were in the upper left quadrant indicating lengthy shutdowns of 200–350 days often result in fewer than 100 deaths per 100 000. In comparison, the baseline no further shutdown scenario (S4, model A) found in the lower right quadrant had 740 deaths per 100 000 with zero extra shutdown days.

Some shutdown strategies failed to control the epidemic as demonstrated in the workplaces-only shutdown model (figure 5c). In this model, shutdown reduced transmission compared to the baseline (table 5), but a considerable proportion of the population acquired infection despite 255 days of shutdowns (20.36% clinical attack rate versus 25.13%). The impact of the shutdowns in this model is not visibly notable (figure 5c), but the epidemic curve is flattened compared to baseline (figure 5a). Other models where shutdowns did not control the epidemic (prevent the clinical attack rate from exceeding 10%) include the 84-day break, 112-day break and schools-only shutdown models (figure 4c,d and figure 5d, respectively).

In a few model realizations (S1 model A, S4 models D and E), the impact of shutting down was not always immediate resulting in a higher peak than observed in March to May 2020 despite shutdown being implemented (figures 2 and 5). While the peaks are smaller compared to the baseline (figure 1a), these models suggest there may be a delay between shutting down and observing a decrease in cases.

## 4. Discussion

This study used an age-stratified agent-based model to explore the impact of shutdowns to control SARS-CoV-2 transmission in Canada. Shutdowns were shown to be an effective tool for reducing and delaying community transmission of SARS-CoV-2 in a largely susceptible population where alternative measures were assumed to be insufficient, and in the absence of vaccines. Effectiveness varied depending on the speed, duration, delay and extensiveness of the shutdown, but in general, there were no models in which shutdowns did not produce a better outcome in terms of the reduced number of cases, hospitalizations and deaths. Globally, there have been many examples showing shutdowns are an effective means for controlling SARS-CoV-2 [1]. Nationally, most Canadians have experienced at least one shutdown during the first wave with success over the summer [2]. However, as we have shown in this paper, shutdowns do not guarantee control of the epidemic.

The main study findings are that there are several shutdown characteristics, health outcomes and socioeconomic costs to balance when selecting a strategy. While some scenarios clearly perform better than others and all shutdowns had some impact on the epidemic, the trade-offs are important to consider based on the local context and situation. Shutdowns were shown to largely delay the epidemic with growing cases, hospitalizations and death rates when the response is slow (either due to a higher threshold for shutting down (S1) or delaying a shutdown (S3)). Whereas the lowest cases, hospitalizations and deaths were seen in scenarios that had higher shutdown days (S1, S2 and S3), which have substantial socioeconomic impact on society and to individuals and families [27,28]. We showed shorter periods of shutdowns and delays between shutdowns when alternative measures are insufficient resulted in less time before a subsequent shutdown is needed to regain control of the epidemic; thus, requiring shutdowns to occur more often and for a longer period than shutting down for a sustained period (S1, S2 and S3). Although there are strong justifications for imposing shutdowns for a limited time only, the lifting of shutdowns too early and without the enhancement of alternative public health measures may result in a longer total shutdown period over the course of the epidemic. Our findings are consistent with the strategies adopted in countries such as New Zealand and Australia who have successfully eliminated COVID-19 cases for months at a time by imposing strict shutdowns without delay when local transmission levels increased beyond low threshold levels, and have applied shutdowns for a sustained period until the local transmission was eliminated [1]. In comparison, the total number of shutdown days these countries have experienced, though lengthier than any single shutdown period imposed in Canada (for example, three and a half months in Melbourne, Australia over summer 2020), have been shorter than the total number of shutdown days experienced by Canadians since March 2020.

The final scenario indicated that the design of partial shutdowns are not equal. Shutdowns of certain sectors were more effective than others, and some combinations performed well when compared to a full

shutdown (S4). The stringency, that is, the sector type and the proportion of those sectors in shutdown, determines the effectiveness of a shutdown. We found workplace-only shutdowns in our models did not contribute significantly to reducing community transmission. This is supported by the fact that the proportion of Canadians who have been teleworking since March 2020 has not changed drastically, suggesting that under the initial rates of teleworking, workplaces are not a main contributor to current transmission [17]. The shutdowns of schools contributed more, but it is mixed-age venues in our models that contribute the most to current transmission, and therefore contributed most in reducing transmission in the models when closed. Similarly, the closures of schools, non-essential businesses and shared public facilities during the first wave contributed to the success in driving community transmission down and their reopening appears to mirror the resurgence in Canada after the summer. At the end of 2020, many provinces in Canada are experiencing a resurgence of SARS-CoV-2 transmission and have imposed a second shutdown but with less success than the first one. This is because the second shutdown has not been as stringent as the initial one and was enacted much later when transmission levels were higher [1,2], thus, despite many provinces have been placed in some form of shutdown for at least a few months and for a longer period than in spring 2020, our cases per capita continue to rise to record levels [1,24]. Additionally, it has been bolstered by declining compliance to personal physical distancing [20–23].

Our study demonstrates the most effective shutdowns were those that were implemented quickly, with minimal delay (S1 and S3), initiated when community transmission is low (S1), sustained for an adequate period (S2) and had a high level of stringency targeting multiple sectors, particularly those driving transmission (S4). Countries that have successfully eliminated SARS-CoV-2 transmission for months at a time have implemented extremely stringent shutdowns during times when community transmission has been low (in some cases when one locally acquired case was identified), applied for months at a time even when only a handful of cases were reported and lifting only when no locally acquired cases had been reported for an extended period [1,29,30]. Shutdowns have widespread negative socioeconomic impacts and should only be used when necessary. To avoid future shutdowns, alternative public health measures should be enhanced during the delay in the epidemic afforded by the closures [5,7]. The precise shutdown strategy will vary depending on the community and household structures, population demographics, background transmission level and human behaviour in adhering to public health measures (which is changing over time and hard to plan for) [20]. The optimal shutdown strategy will also depend on the goal, whether it is to eliminate transmission in the population, to temporarily regain control, to buy time to ramp up other measures, to avoid overwhelming healthcare systems, to minimize the impact on the economy, communities, families and individuals or a combination of these objectives.

Study limitations include uncertainty in some epidemiological characteristics of SARS-CoV-2, such as the role of children and asymptomatic infections in disease transmission or how new SARS-CoV-2 variants might evolve in Canada. We rely on estimates from Canadian data and the literature and assume that these do not change over time. The model does not consider transmission in healthcare and long-term care facility settings because transmissions in these settings vary widely across Canada and are subject to unique infection control and prevention strategies that are not implemented at the population level [31]. While localized outbreaks in healthcare and long-term care facilities may diminish the effectiveness of shutdowns by driving community transmissions higher, the trends in shutdowns identified in this study will still hold true. Further, as most SARS-CoV-2 transmission in Canada is occurring within the community and in households [24], our study remains relevant for Canada. We ran the model for 700 days, which allowed for a certain level of immunity to build up in the population but the level of immunity reached was different for each scenario. The comparisons presented are, therefore, tied to outcomes reported on the model's last day, when immunity levels varied across scenarios. As the impact of interventions will change as herd immunity is reached (S3 scenarios), the results and trends presented should be interpreted with this constraint in mind, that is, the findings presented are estimates of the impact of shutdowns on SARS-CoV-2 transmission in a predominantly naive population and interventions applied to the population at other immunity levels may have a different impact to the scenarios explored. Last, an additional analysis in the electronic supplementary material, appendix shows that health outcomes and shutdown days can be impacted by the duration of model runs, particularly when scenarios have reached different levels of immunity by the end of the model run. Although some differences were observed when exploring longer model runs, there was no deviation in the trends observed in a 700-day model run so the findings presented remain unchanged despite some variations in outcomes.

Our study explores the impact of shutdowns on the resurgence of SARS-CoV-2 transmission in Canada under the assumption that current efforts to control the epidemic remain insufficient in the

absence of a vaccine. We explored and identified shutdown factors that can minimize negative health outcomes, but also showed that there is a trade-off between reducing health outcomes and the number of shutdown days and some strategies may balance both better than others. Given the immense socioeconomic impact of shutdowns, they should be avoided where possible and used wisely as needed to bolster the capacity of other less disruptive public health measures for controlling the epidemic. Although this study is Canadian, our findings are consistent with those observed beyond Canada's border (for example, countries such as Australia and New Zealand who have consistently applied lockdowns when case incidence have been in the single digits) and can be generalized to any country faced with controlling SARS-CoV-2 transmission in their population.

Data accessibility. The model code has previously been published and is available online here: https://nccid.ca/phac-agent-based-model-on-covid-19/. Data for the model have also been made available in the electronic supplementary material.

Authors' contributions. V.N. and N.H.O. contributed to the conceptualization. A.O. and L.A.W. contributed to the data collection. V.N. developed the model and performed the analysis. All of the authors contributed to the interpretation of the results. V.N. drafted the manuscript and all of the authors contributed to the revision of the manuscript. All of the authors gave final approval of the version to be published and agreed to be accountable for all aspects of the work.

Competing interests. We declare we have no competing interests.

Funding. No external funding to declare.

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
