## [Peer Review File · Royal Society Open Science]

Review History

RSOS-210233.R0 (Original submission)

Review form: Reviewer 1

Is the manuscript scientifically sound in its present form?

Yes

Are the interpretations and conclusions justified by the results?

Yes

Is the language acceptable?

Yes

Do you have any ethical concerns with this paper?

No

Have you any concerns about statistical analyses in this paper?

Yes

Recommendation?

Accept with minor revision (please list in comments)

Comments to the Author(s)

The manuscript utilized an Agent-Based Model (published elsewhere) developed for SARS-CoV-2 and tailored to the Canadian population. The model itself is state-of-the-art and well-founded. The work presented in this manuscript leverages the model above and explores different shutdown strategies to control SARS-CoV-2 epidemics under different scenarios. It provides a combination of a well-designed set of experiments that offers useful insights to guide public health measures and aid in decision-making. Its contributions are relevant, timely, and can reduce uncertainty and guesswork when devising such strict interventions as shutdowns.

Questions:

- 1) Page 11, lines 32-40: "shutdown impact was not always immediate..." I agree with the statement, but why is this behaviour observed only when the lower criteria (50 active cases per 100,000) was utilized? I would expect this effect to amplify as the criteria became more permissive. If there are 200 active cases per 100,000, it should take longer to regain control of the epidemic with a shutdown, compared to intervening sooner.
- 2) Page 12, lines 29-30: Herd immunity was mentioned when describing this scenario (S3) but not others. Is there a specific reason for it? Since the model runs for 700 days, a certain level of immunity builds up as the days progress, which is also true for all scenarios. The levels of immunity reached in each simulation will be different for each scenario. This poses a challenge when comparing the different scenarios since earlier and different interventions will impact later ones, precisely this model's point. It would be worth highlighting the absence of herd immunity and the different immunity levels across simulations, more broadly. They are a systematic error inherent to the type of phenomena under investigation, and there is nothing wrong with it.
- 3) In scenario 3, the effects of 84- vs 112-day break is negligible. I wonder if this is related to a lack of resolution of the model caused by the limited number of days in the simulation. The simulation runs for 700 days. Therefore, the longer the break, the less room there is in the simulation for its effects to be observed. Alternatively, the 84-day break may be enough to decrease the number of cases to a level where expanding the break does not significantly impact the outcome. It would be an improvement to expand the discussion in this regard and include a discussion around the attack rate and the number of hospitalizations and deaths. The discussion, as presented, induces the conclusion that longer delays are a good option, but it does not mention that it increases the number of deaths and demands on the healthcare system.
- 4) Table 3, column A: incorrect number of significant digits for Total number of hospitalized cases;
- 5) Table 3, column C: incorrect median or CI for: total number of hosp. cases, total acute hosp. cases, and total cases admitted to ICU.
- 6) Figure 1B: presents the mean value of projections without the associated confidence interval. Although the mean hints that the model describes actual data reasonably well, it is impossible to fully evaluate it without the associated confidence interval.
- 7) Figures 2-5: the authors plot all curves for all 50 realizations in all graphs. This makes it challenging to visualize what the model's output means. I strongly suggest plotting the summary of all simulations, i.e. plotting the average curve for all simulations along with a shaded area

displaying the 95% CI. Such a plot would allow for better visualizations of the results and enable the reader to visualize the different wavelets.

8) Figures 2-5: It would really improve these figures' readability if the prevalent curves were added to these plots. Since the number of active cases is used to trigger shutdowns, it would be very informative to be able to visualize when thresholds are reached or what levels the epidemic reaches when time-constrained interventions are used.

9) Figures 1-5: what does the vertical red-dotted line mean? It is not explained in the caption for the figures, and I could not find it in the text.

10) Figures 1-5: although this is more of a stylistic choice and there is no right or wrong way of presenting these graphs, I would suggest the authors not to use the same Y-scale across figures. Using the same Y-scale in all panels within one figure is good, but across figures does not contribute to the interpretation of the results.

Suggestions:

1) Tables 2 to 5a and 5b could be moved into the appendix and replaced with shorter versions highlighting each experiment's significant findings. It may be possible to combine these tables into one or two smaller summary tables presenting key indicators.

2) I do not think there is anything flawed with the analysis. Still, the number of realizations does not seem adequate (only 50 realizations per scenario). I understand the urgency in reporting these results. Still, I would strongly recommend that the authors present an evaluation (in appendix) of the variability associated with the number of realizations. How does the number of realizations change the conclusions? If the authors have performed such a study, I would strongly recommend its inclusion in the appendix or as a reference. In an Agent-Based Model with such a large number of parameters, it would be surprising if only 50 realizations would be enough to characterize the state space adequately.

3) Page 7, lines 49-54: The model assumes 100% compliance. That sounds quite unrealistic. Even though compliance rates may be high, 100% is most likely unattainable. Even with people trying to be 100% compliant, they probably do not even know how to be 100% compliant. I would suggest relaxing this assumption to more credible numbers (85%-95%).

4) Page 9 line 20: although a large number of reported cases can trigger shutdowns, I would argue that this has changed throughout the epidemic. Some jurisdictions have been focusing more on the economy and only imposing shutdown strategies when the health care system is at risk. Scenario 1 makes sense within a modelling context, but the real-world parallel/justification could be improved. It would be better to say that the model assumed that high rates of identified cases trigger shutdowns, period.

5) Page 19, line 6. I would argue that this study is relevant beyond Canada's borders. As previously mentioned, it aligns with what has been observed elsewhere (Australia and New Zealand). Although the Canadian population was used for the model, its findings should be generalizable.

Review form: Reviewer 2

Is the manuscript scientifically sound in its present form?

Yes

Are the interpretations and conclusions justified by the results?

Yes

Is the language acceptable?

Yes

Do you have any ethical concerns with this paper?

No

Have you any concerns about statistical analyses in this paper?

No

Recommendation?

Accept as is

Comments to the Author(s)

This manuscript is well-written, methodologically well-reasoned and addresses a question of critical importance. I appreciate how well-written the manuscript is, as the authors give a balanced discussion of lockdowns, noting their effectiveness in reducing cases, but also recognizing their substantial socio-economic costs, concluding that lockdowns are appropriate when other NPIs are ineffective, and noting that the time that lockdowns buy, should be used to strengthen other NPIs.

The model parameterization and structure is well-justified and the model is previously published, with accessible documentation. The model is able to to replicate the dynamics of COVID in Canada (Figure 1B).

The question of how to implement lockdowns is still of critical importance for a pandemic response. The authors find that early implementation of lockdowns is beneficial, but there is no clear answer as to what threshold number of active cases should trigger a lockdown, as higher number of lockdown days occur when the threshold for triggering a lockdown is lower (Figure 6). As the author's explain, local considerations likely affect how to best implement lockdowns.

This manuscript is technically sound, and an important and novel contribution.

Decision letter (RSOS-210233.R0)

Dear Dr Ng

On behalf of the Editors, we are pleased to inform you that your Manuscript RSOS-210233 "Modelling the impact of shutdowns on resurging SARS-CoV-2 transmission in Canada" has been accepted for publication in Royal Society Open Science subject to minor revision in accordance with the referees' reports. Please find the referees' comments along with any feedback from the Editors below my signature.

Please submit your revised manuscript and required files (see below) no later than 7 days from today's (ie 15-Mar-2021) date. Note: the ScholarOne system will 'lock' if submission of the revision is attempted 7 or more days after the deadline. If you do not think you will be able to meet this deadline please contact the editorial office immediately.

on behalf of Dr Jianhong Wu (Associate Editor) and Glenn Webb (Subject Editor)
openscience@royalsociety.org

Associate Editor Comments to Author (Dr Jianhong Wu):

Associate Editor: 1

Comments to the Author:

Please try to address one comment on data presentation asap. It advised to have this done within 24 hours.

Reviewer comments to Author:

Reviewer: 1

Comments to the Author(s)

The manuscript utilized an Agent-Based Model (published elsewhere) developed for SARS-CoV-2 and tailored to the Canadian population. The model itself is state-of-the-art and well-founded. The work presented in this manuscript leverages the model above and explores different shutdown strategies to control SARS-CoV-2 epidemics under different scenarios. It provides a combination of a well-designed set of experiments that offers useful insights to guide public health measures and aid in decision-making. Its contributions are relevant, timely, and can reduce uncertainty and guesswork when devising such strict interventions as shutdowns.

Questions:

1) Page 11, lines 32-40: "shutdown impact was not always immediate..." I agree with the statement, but why is this behaviour observed only when the lower criteria (50 active cases per 100,000) was utilized? I would expect this effect to amplify as the criteria became more

permissive. If there are 200 active cases per 100,000, it should take longer to regain control of the epidemic with a shutdown, compared to intervening sooner.

2) Page 12, lines 29-30: Herd immunity was mentioned when describing this scenario (S3) but not others. Is there a specific reason for it? Since the model runs for 700 days, a certain level of immunity builds up as the days progress, which is also true for all scenarios. The levels of immunity reached in each simulation will be different for each scenario. This poses a challenge when comparing the different scenarios since earlier and different interventions will impact later ones, precisely this model's point. It would be worth highlighting the absence of herd immunity and the different immunity levels across simulations, more broadly. They are a systematic error inherent to the type of phenomena under investigation, and there is nothing wrong with it.

3) In scenario 3, the effects of 84- vs 112-day break is negligible. I wonder if this is related to a lack of resolution of the model caused by the limited number of days in the simulation. The simulation runs for 700 days. Therefore, the longer the break, the less room there is in the simulation for its effects to be observed. Alternatively, the 84-day break may be enough to decrease the number of cases to a level where expanding the break does not significantly impact the outcome. It would be an improvement to expand the discussion in this regard and include a discussion around the attack rate and the number of hospitalizations and deaths. The discussion, as presented, induces the conclusion that longer delays are a good option system, but it does not mention that it increases the number of deaths and demands on the healthcare system.

4) Table 3, column A: incorrect number of significant digits for Total number of hospitalized cases;

5) Table 3, column C: incorrect median or CI for: total number of hosp. cases, total acute hosp. cases, and total cases admitted to ICU.

6) Figure 1B: presents the mean value of projections without the associated confidence interval. Although the mean hints that the model describes actual data reasonably well, it is impossible to fully evaluate it without the associated confidence interval.

7) Figures 2-5: the authors plot all curves for all 50 realizations in all graphs. This makes it challenging to visualize what the model's output means. I strongly suggest plotting the summary of all simulations, i.e. plotting the average curve for all simulations along with a shaded area displaying the 95% CI. Such a plot would allow for better visualizations of the results and enable the reader to visualize the different wavelets.

8) Figures 2-5: It would really improve these figures' readability if the prevalent curves were added to these plots. Since the number of active cases is used to trigger shutdowns, it would be very informative to be able to visualize when thresholds are reached or what levels the epidemic reaches when time-constrained interventions are used.

9) Figures 1-5: what does the vertical red-dotted line mean? It is not explained in the caption for the figures, and I could not find it in the text.

10) Figures 1-5: although this is more of a stylistic choice and there is no right or wrong way of presenting these graphs, I would suggest the authors not to use the same Y-scale across figures. Using the same Y-scale in all panels within one figure is good, but across figures does not contribute to the interpretation of the results.

Suggestions:

- 1) Tables 2 to 5a and 5b could be moved into the appendix and replaced with shorter versions highlighting each experiment's significant findings. It may be possible to combine these tables into one or two smaller summary tables presenting key indicators.
- 2) I do not think there is anything flawed with the analysis. Still, the number of realizations does not seem adequate (only 50 realizations per scenario). I understand the urgency in reporting these results. Still, I would strongly recommend that the authors present an evaluation (in appendix) of the variability associated with the number of realizations. How does the number of realizations change the conclusions? If the authors have performed such a study, I would strongly recommend its inclusion in the appendix or as a reference. In an Agent-Based Model with such a large number of parameters, it would be surprising if only 50 realizations would be enough to characterize the state space adequately.
- 3) Page 7, lines 49-54: The model assumes 100% compliance. That sounds quite unrealistic. Even though compliance rates may be high, 100% is most likely unattainable. Even with people trying to be 100% compliant, they probably do not even know how to be 100% compliant. I would suggest relaxing this assumption to more credible numbers (85%-95%).
- 4) Page 9 line 20: although a large number of reported cases can trigger shutdowns, I would argue that this has changed throughout the epidemic. Some jurisdictions have been focusing more on the economy and only imposing shutdown strategies when the health care system is at risk. Scenario 1 makes sense within a modelling context, but the real-world parallel/justification could be improved. It would be better to say that the model assumed that high rates of identified cases trigger shutdowns, period.
- 5) Page 19, line 6. I would argue that this study is relevant beyond Canada's borders. As previously mentioned, it aligns with what has been observed elsewhere (Australia and New Zealand). Although the Canadian population was used for the model, its findings should be generalizable.

Reviewer: 2

Comments to the Author(s)

This manuscript is well-written, methodologically well-reasoned and addresses a question of critical importance. I appreciate how well-written the manuscript is, as the authors give a balanced discussion of lockdowns, noting their effectiveness in reducing cases, but also recognizing their substantial socio-economic costs, concluding that lockdowns are appropriate when other NPIs are ineffective, and noting that the time that lockdowns buy, should be used to strengthen other NPIs.

The model parameterization and structure is well-justified and the model is previously published, with accessible documentation. The model is able to replicate the dynamics of COVID in Canada (Figure 1B).

The question of how to implement lockdowns is still of critical importance for a pandemic response. The authors find that early implementation of lockdowns is beneficial, but there is no clear answer as to what threshold number of active cases should trigger a lockdown, as higher number of lockdown days occur when the threshold for triggering a lockdown is lower (Figure 6). As the author's explain, local considerations likely affect how to best implement lockdowns.

This manuscript is technically sound, and an important and novel contribution.

===PREPARING YOUR MANUSCRIPT===

===PREPARING YOUR REVISION IN SCHOLARONE===

-- If you have uploaded ESM files, please ensure you follow the guidance at <https://royalsociety.org/journals/authors/author-guidelines/#supplementary-material> to include a suitable title and informative caption. An example of appropriate titling and captioning may be found at [https://figshare.com/articles/Table_S2_from_Is_there_a_trade-off_between_peak_performance_and_performance_breadth_across_temperatures_for_aerobic_sc ope_in_teleost_fishes_/3843624](https://figshare.com/articles/Table_S2_from_Is_there_a_trade-off_between_peak_performance_and_performance_breadth_across_temperatures_for_aerobic_scope_in_teleost_fishes_/3843624).

Author's Response to Decision Letter for (RSOS-210233.R0)

See Appendix A.

Decision letter (RSOS-210233.R1)

Dear Dr Ng,

I am pleased to inform you that your manuscript entitled "Modelling the impact of shutdowns on resurging SARS-CoV-2 transmission in Canada" is now accepted for publication in Royal Society Open Science.

COVID-19 rapid publication process:

We are taking steps to expedite the publication of research relevant to the pandemic. If you wish, you can opt to have your paper published as soon as it is ready, rather than waiting for it to be published the scheduled Wednesday.

This means your paper will not be included in the weekly media round-up which the Society sends to journalists ahead of publication. However, it will still appear in the COVID-19 Publishing Collection which journalists will be directed to each week (<https://royalsocietypublishing.org/topic/special-collections/novel-coronavirus-outbreak>).

If you wish to have your paper considered for immediate publication, or to discuss further, please notify openscience_proofs@royalsociety.org and press@royalsociety.org when you respond to this email.

on behalf of Dr Jianhong Wu (Associate Editor) and Glenn Webb (Subject Editor)
openscience@royalsociety.org

Appendix A

Response to reviewers' comments

Reviewer: 1

Comments to the Author(s):

The manuscript utilized an Agent-Based Model (published elsewhere) developed for SARS-CoV-2 and tailored to the Canadian population. The model itself is state-of-the-art and well-founded. The work presented in this manuscript leverages the model above and explores different shutdown strategies to control SARS-CoV-2 epidemics under different scenarios. It provides a combination of a well-designed set of experiments that offers useful insights to guide public health measures and aid in decision-making. Its contributions are relevant, timely, and can reduce uncertainty and guesswork when devising such strict interventions as shutdowns.

Thank you to reviewer #1 for taking the time to review this manuscript and for their positive review. This reviewer has highlighted a number of important issues with the manuscript, particularly the visualisation of the figures and the need for some additional technical exploration of the models presented. We have addressed these issues point-by-point below in either blue or highlighted text.

Questions:

1) Page 11, lines 32-40: "shutdown impact was not always immediate..." I agree with the statement, but why is this behaviour observed only when the lower criteria (50 active cases per 100,000) was utilized? I would expect this effect to amplify as the criteria became more permissive. If there are 200 active cases per 100,000, it should take longer to regain control of the epidemic with a shutdown, compared to intervening sooner.

There are two reasons why we would see a peak in the epidemic when shutting down at 50 active cases per 100,000 compared to when shutting down at 200 active cases per 100,000, these relate primarily to household infections and population-level herd immunity acquired by infection. At 50 active cases per 100,000, while shutdowns will restrict infections between households, there are still some ongoing transmission within household members that shutdowns would not be able to control (because agents do not physically distance from household members, as would be expected in real life). At 200 active cases per 100,000, a larger proportion of the population has already succumbed to infection, including many household members, thus shutting down when active cases reach 200 active cases per 100,000 (which is equivalent to approximately 75,000 active cases for the Canadian population) would result in shutting down when a high level of herd immunity has already been acquired by the population. In our modelling results where shutdown is implemented at 100 active cases per 100,000, we also observe delays in the impacts of shutdowns (previous Figure 5 showed this but the updated figures now do not show the outlier model runs due to model runs being presented as a shaded 95% credible interval). Similarly, we have observed this (but rarely) when 200 active cases per 100,000 is used as the trigger for shutting down but this was not observed in the 50 simulation runs that we ran and present in this manuscript. We address the extreme model outputs from larger model runs in an updated section of the

Appendix which is a response to one of the review's comments below on whether 50 runs covers the full range of stochastic outputs that can be observed given the large number of parameters in the model. For this comment, to address the issue of why the impact of shutdowns were not always observed when the threshold was 50 active cases per 100,000, we have added the following section to the revised manuscript:

Pages 10 and 11, lines 207 to 219: "In a few model runs when shutting down at 50 active cases per 100,000 (Figure 2, panel A), the shutdown impact was not always immediate resulting in a larger epidemic peak despite shutdown being implemented (though this was noticeably smaller compared to the baseline – Figure 1, panel A). This phenomenon was also observed in scenarios when shutting down at 100 active cases per 100,000 (not visible as the few extreme runs are not captured in the 95% credible intervals) and occasionally at 150 or 200 active cases per 100,000. This is due to shutdowns having a direct impact on inter-household transmission but not within-household transmission; the latter will continue when shutdowns are implemented because household members do not physically distance from each other. Therefore, at 50 active cases per 100,000, a larger proportion of household members remain susceptible to infection whereas at 150 or 200 active cases per 100,000, many household members have succumbed to infection and there is a high level of herd immunity at the population level."

2) Page 12, lines 29-30: Herd immunity was mentioned when describing this scenario (S3) but not others. Is there a specific reason for it? Since the model runs for 700 days, a certain level of immunity builds up as the days progress, which is also true for all scenarios. The levels of immunity reached in each simulation will be different for each scenario. This poses a challenge when comparing the different scenarios since earlier and different interventions will impact later ones, precisely this model's point. It would be worth highlighting the absence of herd immunity and the different immunity levels across simulations, more broadly. They are a systematic error inherent to the type of phenomena under investigation, and there is nothing wrong with it.

Yes, we had only referenced herd immunity when describing scenario S3 because this is the only scenario in which some models reach herd immunity (112-day break between shutdowns and 84-day break between shutdowns (almost for the latter)). However, the reviewer brings up a good point about the varying levels of immunity reached in each simulation. We have now updated the manuscript to discuss the impact of immunity in some of the other scenarios. Additionally, we have addressed the limitations of the absence of herd immunity reached by a 700-day model run and the challenge this poses to interpreting model results.

For scenario 1, page 10, lines 204-207: "Prevalent cases are on average three-fold higher at the beginning of the epidemic when shutting down at 200 active cases compared to shutting down at 50 active cases per 100,000. This difference gradually drops to two-fold as immunity level increases in the higher threshold scenarios (Figure 2)."

For Scenario 2, page 12, lines 238-240: "In all four scenarios, a decrease in prevalent cases is observed with each subsequent shutdown indicating a gradual increase in immunity level in the population."

We have also included the following in the limitations section to address the limitations of the absence of herd immunity reached by day 700 in the majority of the scenarios:

Page 19, lines 399-412 “We ran the model for 700 days, which allowed for a certain level of immunity to build up in the population but the level of immunity reached was different for each scenario. The comparisons presented are, therefore, tied to outcomes reported on the model’s last day, when immunity levels varied across scenarios. As the impact of interventions will change as herd immunity is reached (S3 scenarios), the results and trends presented should be interpreted with this constraint in mind, that is, the findings presented are estimates of the impact of shutdowns on SARS-CoV-2 transmission in a predominantly naïve population and interventions applied to the population at other immunity levels may have a different impact to the scenarios explored. An additional analysis in the Appendix show that health outcomes and shutdown days can be impacted by the duration of model runs, particularly when scenarios have reached different levels of immunity by the end of the model run. Although some differences were observed when exploring longer model runs, there was no deviation in the trends observed in a 700-day run so the findings presented remain unchanged despite some variations in outcomes.”

Last, we included an additional analysis and discussion on the impact of immunity levels and herd immunity on the interpretation of model outputs in the Appendix (pages 17-22).

3) In scenario 3, the effects of 84- vs 112-day break is negligible. I wonder if this is related to a lack of resolution of the model caused by the limited number of days in the simulation. The simulation runs for 700 days. Therefore, the longer the break, the less room there is in the simulation for its effects to be observed. Alternatively, the 84-day break may be enough to decrease the number of cases to a level where expanding the break does not significantly impact the outcome. It would be an improvement to expand the discussion in this regard and include a discussion around the attack rate and the number of hospitalizations and deaths. The discussion, as presented, induces the conclusion that longer delays are a good option, but it does not mention that it increases the number of deaths and demands on the healthcare system.

To address this, we have included two additional sections in the Appendix to explore the impact of a longer model run time of 1096 days (3 years) on two comparisons, one between a 28-day shutdown and 70-day shutdown and one between an 84-day break and a 112-day break (longer model run time of 1096 in addition to 200 model realizations). We do find some differences and we present and discuss our findings in the Appendix (pages 17-18, lines 324-355).

In addition, we have addressed this issue in the results section for Scenario 3:

Page 12, lines 247-250: “Although it appears the effects of an 84-day break and 112-day break between shutdowns is negligible, further analysis from a longer model run showed that a significantly higher number of shutdown days is needed to control the epidemic in the 84-day break scenario (Appendix).”

Last, the following sentence in the limitation section referring to the analysis in the Appendix:

Page 19, lines 407-412 “Last, an additional analysis in the Appendix show that health outcomes and shutdown days can be impacted by the duration of model runs, particularly when scenarios have reached different levels of immunity by the end of the model run. Although some difference were observed when exploring longer model runs, there was no deviation in the trends observed in a 700-day model run so the findings presented remain unchanged despite some variations in outcomes.”

To address the point on expanding the discussion to include the number of deaths and demands on the healthcare system, and to highlight that longer delays are actually not a better option, we have included the following in the section discussing Scenario 3 results:

Page 12, lines 258-263: “Delaying a shutdown when a large proportion of the population remains naïve can result in up to 10 times the number of prevalent clinical cases at peak compared to a no-delay approach (~800 per 100,000 with a 112-day break compared to ~80 per 100,000 with a 28-day break), thus overwhelming the healthcare system (Figure 4). Total hospitalizations increased by 3-fold in the 112-day break scenario compared to the 28-day break scenario while deaths increased by 6-fold.”

4) Table 3, column A: incorrect number of significant digits for Total number of hospitalized cases.

We have updated the total number of hospitalized cases in Table 3, column A so that hospitalizations are rounded up and presented in full numbers, consistent with the other columns in the same table. As we present median values across 50 model simulations, values of 0.5 can appear in median values. The value has been updated from 519.5 total hospitalized cases per 100,000 to 520 total hospitalized cases per 100,000.

5) Table 3, column C: incorrect median or CI for: total number of hosp. cases, total acute hosp. cases, and total cases admitted to ICU.

Thank you to the reviewer for their sharp eye, in addition to correcting the number of hospitalised cases, acute hospitalised cases and cases admitted to the ICU, we found a small error in the total cases, clinical cases, asymptomatic cases and death outputs due to a coding error in our extraction code. The error did not impact the credible intervals and we have updated the correct median values in Table 3 as follows:

Number of shutdown days (from Sep 8, 2020 to Jan 7, 2022)	211 (171-256)	218 (185-256)	229 (224-283)
Number of days contact tracing/quarantine paused (full model run)	401 (321-484)	340 (253-382)	293 (218-363)
Total cases (clinical and asymptomatic) per 100,000	8461 (7298-9991)	7580 (6633-8368)	6933 22 (5623-7924)
Total clinical cases per 100,000	5180 (4528-6155)	4652 (4081-5063)	4227 55 (3499-4844)
Total asymptomatic cases per 100,000	3271 (2791-3836)	2943 (2540-3305)	2662 77 (2124-3090)
Total hospitalized cases (acute and ICU) per 100,000	520 19.5 (462-648)	466 (391-538)	431 44 (355-492)
Total acute hospitalized cases per 100,000	383 (343-475)	341 (288-412)	317 33 (259-367)
Total cases admitted into the ICU per 100,000	135 (115-168)	117 (95-141)	108 55 (80-133)
Total deaths per 100,000	74 (57-90)	65 (50-85)	59 60 (40-74)

6) Figure 1B: presents the mean value of projections without the associated confidence interval. Although the mean hints that the model describes actual data reasonably well, it is impossible to fully evaluate it without the associated confidence interval.

We have updated all of our figures to now include the 95% credible intervals with a solid black line representing the median value across 50 model simulations. For Figure 1B, we now present the number of observed cases for comparison with the median value of projections and the associated 95% credible interval across simulations. We have also updated Figure S2 in the appendix and have included the following text to explain the model fit in terms of the 95% credible interval. We note that we had previously incorrectly referred to the mean value of projections rather than the median value of projections. This has now been corrected in the manuscript and the Appendix. In the Appendix, pages 14 to 15, lines 258-261, we have included this text:

“While the 95% credible interval may be wide, this is an inherent feature in stochastic models where by chance under the same set of conditions, an epidemic may or may not emerge. We selected the median value to fit our model to as this is the best metric to reduce the influence of outlier model runs.”

7) Figures 2-5: the authors plot all curves for all 50 realizations in all graphs. This makes it challenging to visualize what the model's output means. I strongly suggest plotting the summary of all simulations, i.e. plotting the average curve for all simulations along with a shaded area displaying the 95% CI. Such a plot would allow for better visualizations of the results and enable the reader to visualize the different wavelets.

We agree with the reviewer’s comments and have now plotted each figure to show the model’s median curve across simulations with a shaded area displaying the 95% credible interval. As the reviewer has suggested, this better visualises the results and enables the reader to see the wavelets created by repetitious shutdowns. One detail that is lost is that the wavelets are not always moving in sync across realizations so the shaded 95% credible interval masks this feature, however, the figures are clearer by presenting the simulations in this manner so we have accepted this suggestion and updated all plots in Figures 2 to 5 accordingly.

8) Figures 2-5: It would really improve these figures' readability if the prevalent curves were added to these plots. Since the number of active cases is used to trigger shutdowns, it would be very informative to be able to visualize when thresholds are reached or what levels the epidemic reaches when time-constrained interventions are used.

We have now included a red dash line representing the median prevalent active cases across 50 model realizations in Figures 2 to 5. This has added much value visually to the figures and we thank the review for this suggestion. In addition, we have added in the following text to refer to the prevalent curves added to the plots (yellow highlight represents new text):

For scenario 1, page 10, lines 204-207: “Prevalent cases are on average three-fold higher at the beginning of the epidemic when shutting down at 200 active cases compared to shutting down at 50 active cases per 100,000. This difference gradually drops to two-fold as immunity level increases in the higher threshold scenarios (Figure 2).”

For Scenario 2, page 11, lines 222-227: “When public health interventions are insufficient, shorter shutdowns are triggered more frequently: seven complete cycles of shutdowns were observed for the 28-day shutdown model; five cycles of shutdowns for the 42-day model; four cycles of shutdowns for the 56-day model and three to four cycles of shutdowns for the 70-day model when the trigger for shutting down was set to 100 active cases per 100,000 (Figure 3). These cycles are observed in both the median incident and prevalent clinical cases.”

Also for Scenario 2, pages 11 and 12, lines 234-240: Unlike S1, the initial prevalent clinical case count at which shutdowns were triggered were similar across scenarios in S2. Shorter period of shutdowns, when alternative public health measures are insufficient, resulted in more cases, hospitalizations and deaths compared to sustained periods of shutdowns in S2. Comparison of the duration of shutdowns (S2) and the timing of the shutdown (S1) show a higher impact of timing quick re-implementation of the shutdowns. In all four scenarios, a decrease in prevalent cases is observed with each subsequent shutdown indicating a gradual increase in immunity level in the population.”

For Scenario 3, pages 12 and 13, lines 258-263: “Delaying a shutdown when a large proportion of the population remains naïve can result in up to 10 times the number of prevalent clinical cases at peak compared to a no-delay approach (~800 per 100,000 with a 112-day break compared to ~80 per 100,000 with a 28-day break), thus overwhelming the healthcare system (Figure 4)”.

9) Figures 1-5: what does the vertical red-dotted line mean? It is not explained in the caption for the figures, and I could not find it in the text.

We have removed the vertical red-dotted lines from all figures to help make the figures more visually appealing and less cluttered. These had previously referred to the point in time with physical distancing compliance shifted in the population, in line with Canadian surveys. In addition, we have removed the three coloured phases of physical distancing to make the figures more visually appealing with less distraction.

10) Figures 1-5: although this is more of a stylistic choice and there is no right or wrong way of presenting these graphs, I would suggest the authors not to use the same Y-scale across figures. Using the same Y-scale in all panels within one figure is good, but across figures does not contribute to the interpretation of the results.

Yes, we agree with the reviewer on this and we have updated every plot in Figures 1 to 5 so that the Y-scale across separate figures are not consistent, to allow for a better interpretation of the results. For the Y-scale in all panels within one figure, we have kept the Y-scales across the figure consistent to allow for comparison.

Suggestions:

1) Tables 2 to 5a and 5b could be moved into the appendix and replaced with shorter versions highlighting each experiment's significant findings. It may be possible to combine these tables into one or two smaller summary tables presenting key indicators.

We have now removed the number of days contact tracing has been paused and the number of infections occurring in schools, workplaces, mixed age venues and in the household as these were not referred to in the manuscript. We have however left the tables in the manuscript as they are referenced extensively in the results section of the manuscript. The removed parameters can all be found in the supplementary table that includes all model outputs for all scenarios and across all ten age groups so the reader still has access to this information in the appendix. However, we feel the updated and more concise tables do add value to the results section but we would be happy to work with the Editor to present them in a more suitable format.

2) I do not think there is anything flawed with the analysis. Still, the number of realizations does not seem adequate (only 50 realizations per scenario). I understand the urgency in reporting these results. Still, I would strongly recommend that the authors present an evaluation (in appendix) of the variability associated with the number of realizations. How does the number of realizations change the conclusions? If the authors have performed such a study, I would strongly recommend its inclusion in the appendix or as a reference. In an Agent-Based Model with such a large number of parameters, it would be surprising if only 50 realizations would be enough to characterize the state space adequately.

We have included in the Appendix (pages 23-25) an exploration of two different models (the baseline and the 70-day shutdown scenario) and compared the differences between a 50-realization model run and a 200-realization model run across 3 years. We found that a 200-realization model provides more stable model outputs but a 50-realization model was sufficient to analyse the general trends across and between the presented scenarios in this study. This could be because we explore scenarios that are substantially different to each other. If we were to explore less variation in our scenarios, for example, a 60-day shutdown versus a 70-day shutdown, utilising additional model realizations would be beneficial and allow for distinct and stable model outputs.

3) Page 7, lines 49-54: The model assumes 100% compliance. That sounds quite unrealistic. Even though compliance rates may be high, 100% is most likely unattainable. Even with people trying to be 100% compliant, they probably do not even know how to be 100% compliant. I would suggest relaxing this assumption to more credible numbers (85%-95%).

While we agree with the reviewer's comment here on relaxing compliance to physical distancing during the early phase the epidemic, this was the only suggestion we did not address. We did this for a number of reasons:

1. Although we assumed a 100% compliance in physical distancing outside of the household, this assumption was not applied for within the household; which is where a lot of contacts occur for individuals not necessarily living with their family members at a time when most places were shut down and inaccessible, i.e. the model still accounts for social gatherings in the private setting with no physical distancing applied
2. By July 1 2020 and until the model end, January 2022, we assume that compliance changes to the following by age group: contact rates were reduced to only 55% with compliance shifting by age group: 0-4 (33%), 5-9 (33%), 11-14 (33%), 15-19 (33%), adult1 (50%), adult2 (75%), adult3 (85%), senior1 (90%), senior2 (95%), elderly (95%) (21, 23). As such, the period of time in which we assume 100% compliance is only 10 weeks for a model run for 2 years
3. We based our compliance values on a University of Guelph study funded by PHAC. While their data are not publically available yet, they have provided us with data to help us parameterise our model. Their survey of 5,000 individuals indicate that during the early phase of the epidemic, quite a large proportion of individuals were compliant with physical distancing and reducing their contacts from pre-COVID rates (upward of approximately 95%).

However, we are constantly updating our model and we will take this suggestion into consideration for future model fitting.

We have also included the following sentence in the Appendix (page 14, lines 235-239): “Although we apply 100% compliance in physical distancing in Phase 1 and 2 (from March 16 to June 30, 2020), physical distancing is not adhered to within the household. During this period when a large proportion of places were shut down, contacts were shifted to household members, which will account for social gatherings in the private setting with no physical distancing applied. “

4) Page 9 line 20: although a large number of reported cases can trigger shutdowns, I would argue that this has changed throughout the epidemic. Some jurisdictions have been focusing more on the economy and only imposing shutdown strategies when the health care system is at risk. Scenario 1 makes sense within a modelling context, but the real-world parallel/justification could be improved. It would be better to say that the model assumed that high rates of identified cases trigger shutdowns, period.

We have updated this paragraph in the revised manuscript as follows:

Page 8, lines 149-156: “Scenario 1 (S1) explored the speed of shutdowns when the following triggers were reached, reflecting some Canadian thresholds that have been set to trigger shutdowns (25-27): (A) 50 active cases per 100,000 (fast response), (B) 100 active cases per 100,000, (C) 150 active cases per 100,000 and (D) 200 active cases per 100,000 (slow response). Active cases in the model represented the total number of cases that were symptomatic or hospitalized on a given day, i.e., clinical cases once symptoms have begun. These thresholds were selected to reflect thresholds used by some provinces in Canada while recognising some jurisdictions use healthcare system capacity as a trigger for shutdowns, which are also considered jointly with economic impacts.”

5) Page 19, line 6. I would argue that this study is relevant beyond Canada's borders. As previously mentioned, it aligns with what has been observed elsewhere (Australia and New Zealand). Although the Canadian population was used for the model, its findings should be generalizable.

We agree with this comment and we have included the following sentence in the final concluding paragraph of the revised manuscript:

Page 20, lines 420-423: “Although this study is Canadian, our findings are consistent with those observed beyond Canada’s border (for example, countries such as Australia and New Zealand who have consistently applied lockdowns when case incidence have been in the single digits) and can be generalised to any country faced with controlling SARS-CoV-2 transmission in their population.”

Reviewer: 2

Comments to the Author(s):

This manuscript is well-written, methodologically well-reasoned and addresses a question of critical importance. I appreciate how well-written the manuscript is, as the authors give a balanced discussion of lockdowns, noting their effectiveness in reducing cases, but also recognizing their substantial socio-economic costs, concluding that lockdowns are appropriate when other NPIs are ineffective, and noting that the time that lockdowns buy, should be used to strengthen other NPIs.

The model parameterization and structure is well-justified and the model is previously published, with accessible documentation. The model is able to replicate the dynamics of COVID in Canada (Figure 1B).

The question of how to implement lockdowns is still of critical importance for a pandemic response. The authors find that early implementation of lockdowns is beneficial, but there is no clear answer as to what threshold number of active cases should trigger a lockdown, as higher number of lockdown days occur when the threshold for triggering a lockdown is lower (Figure 6). As the author's explain, local considerations likely affect how to best implement lockdowns.

This manuscript is technically sound, and an important and novel contribution.

We would like to thank reviewer #2 for their time in reviewing our manuscript and for their positive review of this study.